# Potential benefits of advanced chelate-based trace minerals in improving bone mineralization, antioxidant status, immunity, and gene expression modulation in heat-stressed broilers

Taher Mohammadizad[1], Kamran Taherpour[1☯*], Hossein Ali Ghasemi[2☯*], Hassan Shirzadi[1], Fatemeh Tavakolinasab[1], Mohammad Hassan Nazaran[3]

1 Faculty of Agriculture, Department of Animal Science, Ilam University, Ilam, Iran, 2 Faculty of Agriculture and Environment, Department of Animal Science, Arak University, Arak, Iran, 3 Department of Research and Development, Sodour Ahrar Shargh Company, Tehran, Iran

☯ These authors contributed equally to this work.
* k.taherpour@ilam.ac.ir (KT); h-ghasemi@araku.ac.ir (HAG)

## Abstract

Organic sources of trace minerals (TM) in broiler diets are more bioavailable and stable than inorganic sources, making them particularly beneficial during challenging periods such as heat stress (HS) conditions. A 42-d study investigated the effects of using advanced chelate technology-based TM (ACTM) or adding varying amounts of ACTM to broiler diets during HS conditions. The study involved 672 male broiler chickens in 7 treatment groups, including a thermoneutral control (TNC) group and six HS treatments. There were 8 replicate pens per treatment and 12 birds per replicate. The six HS treatments included birds exposed to a cyclic HS environment (34°C) for 8 h and were as follows: HSC, which consisted of the same basal diet with the recommended ITM levels; ACTM50 and ACTM100, which replaced the basal diet with 50% and 100% ACTM instead of ITM; ITM+ACTM12.5 and ITM+ACTM25, which involved adding extra ACTM to the ITM basal diet at 12.5% and 25%, respectively; and ITM125, which used 125% of the recommended levels of ITM in the basal diet. Compared with the HSC treatment, the TNC, ACTM100, and ITM+ACTM25 treatments resulted in increased ($P < 0.05$) body weight; tibia weight; tibia ash, phosphorus, iron, and manganese contents; secondary antibody titers; and serum TAC and SOD values but decreased ($P < 0.05$) serum MDA concentrations and the expression levels of the hepatic genes IL-1β, IL-6, and INF-γ. The TNC and ACTM100 groups also showed greater ($P < 0.05$) feed efficiency, tibia length, tibia zinc content, and hepatic SOD1 expression but exhibited reduced ($P < 0.05$) hepatic NF-kB expression. Significant increases ($P < 0.05$) in primary anti-NDV titers, serum GPx1 activity, and Nrf2 and GPx1 gene expression levels were also detected in the ACTM100, ITM+ACTM12.5, and ITM+ACTM25 groups. In conclusion, the findings suggest that replacing ITM with ACTM or adding ACTM to ITM diets, especially at a 25% higher dose, can

**Data Availability Statement:** All relevant data are within the manuscript and its Supporting Information files.

**Funding:** The author(s) received no specific funding for this work.

**Competing interests:** The authors have declared that no competing interests exist.

effectively protect broilers from heat stress by promoting growth, reducing inflammation, and increasing the expression of antioxidant proteins.

## Introduction

Heat stress (HS) is a common environmental stressor that can significantly affect physiological and metabolic processes in animals, including poultry. The breeding of broiler chickens with higher metabolic rates has made them more susceptible to HS [1,2]. Previous studies have shown that elevated temperatures can harm the health and well-being of birds by compromising their immune function, increasing mortality rates, and reducing feed efficiency and growth performance [3,4]. HS-induced oxidative stress is commonly associated with inflammatory pathway activation, resulting in tissue damage in birds [5]. Previous studies have shown that heat stress can activate the nuclear factor kappa B (NF-kB) signaling pathway, leading to inflammation in organs such as the liver [6] and gut [7]. The activation of this molecule is important for regulating inflammatory mediators such as interleukin-1 beta (IL-1β), interleukin-6 (IL-6), and interferon-gamma (IFN-γ), which affect the initiation and progression of inflammation [8]. Previous research has also identified the nuclear factor erythroid 2-related factor 2 (Nrf2) signaling pathway as a potential mechanism for regulating oxidative stress induced by HS [9]. In stress-induced conditions, Nrf2 moves to the nucleus and triggers the activation of protective genes, including those encoding antioxidants such as superoxide dismutase (SOD), glutathione peroxidase (GPx), and catalase (CAT) [10]. Therefore, there has been an increasing emphasis on the importance of antioxidants in minimizing oxidative stress and its associated physiological issues. Trace minerals (TM) such as zinc (Zn), copper (Cu), manganese (Mn), iron (Fe), and selenium (Se) are potential external antioxidants for reducing oxidative stress [11,12]. Furthermore, iodine (I) plays a vital role in the biological activity of thyroid hormones, while chromium (Cr) functions as an insulin cofactor, and both may also possess antioxidant properties [13,14].

Interactions between minerals and feed components can decrease their bioavailability and contribute to deficiencies [15]. Organic TM, found in chelated or complexed forms, are bound to organic compounds such as amino acids, proteins, or organic acids [16–18]. These minerals have a more stable structure, aiding digestion and absorption in the intestinal tract [19]. Therefore, the inclusion of organic minerals in animal diets, along with appropriate levels, may be justified due to the excessive excretion of these minerals during periods of HS. Although a direct correlation between the supplementation of organic TM in broiler diets and enhanced broiler performance has not been established [20], it has been observed that the inclusion of a single organic TM or a combination of organic TM in diets can have beneficial effects on various physiological processes, including antioxidant defense, immune response, and inflammation regulation. For example, supplementation with organic TM has been shown to lead to a reduction in lipid peroxidation [21], an increase in the levels of immunoglobulins [22], an increase in the activity of SOD and GPx [23], and an increase in the expression levels of anti-inflammatory factors [24].

Advanced chelation technology is a new method for synthesizing structures in different fields [25]. In this method, through self-assembly polymerization of organic acids, various structures are synthesized, the efficacy of which has been proven in numerous studies [26–32]. Through these experiments, the assessment of various physiological functions revealed that structures based on advanced chelation technology can effectively control a range of biological

processes in plants, animals, or humans to maximize outcomes, leading to the term "metabolism optimizer behavior" for these effects. Previous studies have demonstrated positive outcomes when using organic acid-chelated TM in broiler chickens, laying hens, and turkeys [33–35]. However, the effects of advanced chelate-based TM on the bone health status, immunity, and antioxidant indicators of broilers exposed to HS are still unknown. Therefore, this study aimed to evaluate the effectiveness of an advanced chelated form of TM (**ACTM**) in heat-stressed broiler chickens by substituting or supplementing inorganic TM with this chelated form at different levels. The focus was on assessing the effects on bone quality and mineralization, antibody responses, hepatic expression levels of NF-kB-dominated pathways, blood antioxidant status, and hepatic expression levels of Nrf2-dominated pathways in heat-stressed broilers.

## Materials and methods

### Preparation of chelated TM

The ACTM product (Bonzachicken) used in this study was produced using a self-assembly method outlined in a patent (US8288587B2). The structural stability of the product is due to the encapsulation of core ions within the polymerized chelating agent structure. The stability of this structure across different pH values and concentrations of various TMs was investigated in previous studies [33,35]. BET analysis was used to determine the surface areas for advanced chelate crystallinity (S1 Fig). The BET surface areas calculated from the nitrogen isotherms agree well with the accessible surface areas obtained directly from the crystal structures in a geometric fashion.

### Ethics statement

In this research study, the Animal Ethics Committee of Ilam University approved the use of animal trials (approval number 80–4188), which were conducted in accordance with the regulations outlined in the Guide for the Care and Use of Experimental Animals. All the procedures involving animals in this study adhere to the ARRIVE guidelines 2.0. The well-being and welfare of the birds were prioritized during the husbandry and euthanasia processes.

### Birds, diets, and management

A 42-day study involved a total of 672 one-day-old male broiler chickens, which were randomly assigned to 7 groups. Each group had 8 replications, and each pen housed 12 birds. The birds were provided with unlimited access to mesh food and water. In this study, broilers were assigned to seven diets starting from day 0. The treatment groups were as follows: TNC, basal diet with recommended ITM levels, reared under thermoneutral conditions; HSC, basal diet with recommended ITM levels, reared under HS conditions; ACTM50, basal diet with 50% match levels instead of ITM, reared under HS conditions; ACTM100, basal diet with 100% match levels instead of ITM, reared under HS conditions; ITM+ACTM12.5, basal diet with ACTM added at 12.5% above recommended levels, reared under HS conditions; ITM+-ACTM25, basal diet with ACTM added at 25% above recommended levels, reared under HS conditions; and ITM125, basal diet with 125% of recommended ITM levels, reared under HS conditions.

The amount of inorganic TM premix added was 2.5 g/kg for the TNC and HSC treatments and 3.125 g/kg for the ITM125 treatment. The inorganic TM premixes used in the TNC, HSC, and ITM125 treatments were formulated using ferrous sulfate, zinc sulfate, manganese sulfate, copper sulfate, sodium selenite, potassium iodide, and potassium dichromate. Different

dosages of the Bonzachicken supplement were administered to achieve specific levels of ACTM, with the ACTM50 treatment receiving 1 g/kg and the ACTM100 treatment receiving 2 g/kg. ACTM was supplemented onto the ITM diet at the following two concentrations: 250 mg/kg in the ITM+ACTM12.5 group and 500 mg/kg in the ITM+ACTM25 group. The diets used in the study followed the nutrient recommendations outlined in the Ross 308 breeding guide for the starter, grower, and finisher stages (Table 1), with the exception of TM, which was included as part of the experimental treatments. S1 Table displays the composition of TM in the different experimental diets.

## Environmental conditions

The experiment involved two environment-controlled rooms with independent temperature controls. Each room had a thermostat-controlled portable electric heater to regulate the temperature. Humidity levels were controlled using a humidifier with a 5 L capacity, maintaining

**Table 1. Ingredient composition and calculated nutrient contents of basal diets (as-fed basis).**

| Item | Starter (d 0 to 10) | Grower (d 10 to 24) | Finisher (d 24 to 42) |
|---|---|---|---|
| **Ingredients (%)** | | | |
| **Corn** | 55.46 | 57.72 | 61.35 |
| **Soybean meal, 44%** | 32.18 | 30.15 | 26.71 |
| **Corn gluten meal, 60%** | 5.25 | 4.25 | 3.1 |
| **Soybean oil** | 2.2 | 3.5 | 4.8 |
| **Dicalcium phosphate** | 1.95 | 1.71 | 1.5 |
| **Limestone** | 1.16 | 1.07 | 1 |
| **Salt (NaCl)** | 0.22 | 0.23 | 0.2 |
| **Sodium bicarbonate** | 0.11 | 0.1 | 0.14 |
| **Vitamin premix[1]** | 0.25 | 0.25 | 0.25 |
| **Trace mineral premix[2]** | 0–0.25 | 0–0.25 | 0–0.25 |
| **DL-Methionine** | 0.26 | 0.23 | 0.22 |
| **L-Lysine HCl** | 0.43 | 0.32 | 0.28 |
| **L-Threonine** | 0.21 | 0.15 | 0.13 |
| **Building sand** | 0–0.32 | 0–0.32 | 0–0.32 |
| **Total** | 100 | 100 | 100 |
| **Calculated nutritive value** | | | |
| **Metabolizable energy, kcal/kg** | 3000 | 3100 | 3200 |
| **Crude protein (%)** | 23.0 | 21.5 | 19.5 |
| **Calcium (%)** | 0.96 | 0.87 | 0.79 |
| **Nonphytate phosphorus (%)** | 0.48 | 0.44 | 0.40 |
| **Sodium (%)** | 0.16 | 0.16 | 0.16 |
| **Digestible lysine (%)** | 1.28 | 1.15 | 1.03 |
| **Digestible methionine (%)** | 0.62 | 0.55 | 0.51 |
| **Digestible methionine + cysteine (%)** | 0.95 | 0.87 | 0.80 |
| **Digestible threonine (%)** | 0.86 | 0.77 | 0.69 |
| **DEB[3], mEq/kg** | 250 | 240 | 230 |

[1]Supplied per kg diet: 18 mg retinol, 4 mg cholecalciferol, 36 mg a-tocopherol acetate, 2 mg vitamin $K_3$, 1.75 mg vitamin $B_1$, 6.6 mg vitamin $B_2$, 9.8 mg niacin, 29.65 mg pantothenic acid, 2.94 mg vitamin $B_6$, 1 mg folic acid, 0.015 mg vitamin $B_{12}$, 0.1 mg biotin, 250 mg choline chloride and 1 mg ethoxyquin.

[2]The trace mineral (TM) supplementation were referred to our experimental design. The TM premixes were added in place of the building sand that is used as inert filler to adjust the formulation.

[3]DEB (dietary electrolyte balance) = ($Na^+$, mEq/kg + $K^+$, mEq/kg)–$CL^-$, mEq/kg.

a range of 55% ± 5%. The temperature of the chickens was closely monitored throughout the experiment. Initially, the chickens were kept at a temperature range of 33 to 34˚C for the first 3 days. After that, the temperature in the TNC group was gradually reduced by 3˚C per week until it reached a final temperature of 22–23˚C. In contrast, the temperature in the HS groups remained consistent at 33 to 34˚C for 8 hours (10:00–18:00) before dropping to match the TNC group's temperature for the remaining 16 hours daily. To ensure proper ventilation, a mechanical ventilation system was used. This system employed a continuous-flow, pressure-controlled ventilator to effectively circulate air throughout the rooms. The lighting schedule involved a 24-hour period of uninterrupted light from day 0 to day 3, followed by a pattern of 23 h of light and 1 h of darkness from day 3 until the conclusion of the experiment.

## Productive traits and sampling

During the entire duration of the experiment, the amount of feed consumed and the weight of the chickens were recorded, and the feed efficiency (body weight gain: feed intake) was calculated. The daily mortality rate of each pen was documented and utilized to modify the performance metrics. The uniformity rate was determined by calculating the coefficient of variation of the individual body weights at 42 days of age [36].

At the end of the experiment (day 42), one chicken per replicate pen (8 per treatment) were selected based on their average body weight. Blood samples were collected from the wing veins of each bird to obtain serum samples for blood antioxidant analysis. The samples were then centrifuged at 2,500 g for 15 minutes at 4˚C. After blood collection, the birds were humanely euthanized via cervical dislocation by a trained animal technician. Thereafter, birds were dissected, and the tibiae on both sides were carefully separated and cleaned to remove any remaining tissues. The right tibia was used for morphological assessment, while the left tibia was wrapped in gauze, soaked in isotonic saline solution, and frozen at -20˚C for future examination and analysis. Liver samples were obtained, and one lobe from each sample was rinsed with a 0.90% w/v NaCl solution, subsequently flash-frozen in liquid nitrogen, and finally stored at -80˚C for gene expression analysis.

## Blood antioxidant parameters

Spectrophotometric methods were used to measure the enzyme activities of GPx, SOD, and CAT, as well as the levels of total antioxidant capacity (**TAC**) and malondialdehyde (**MDA**) in the serum. The measurements were conducted using commercial kits, according to the manufacturer's instructions. The TAC assay was conducted utilizing a TAC assay kit (Randox Laboratories Ltd., Crumlin, UK). The GPx, SOD, CAT, and MDA assays were performed using the corresponding assay kits (Cayman Chemical Co., Ann Arbor, MI, USA).

## Tibial bone parameters

In this study, a digital caliper was used to measure the length and width of the right tibia. An Instron instrument (model 5500, Instron Corp., Canton, MA) was used to test the breaking strength of the tibias. The left tibia bone samples were dried and then exposed to 600˚C for 6 h to undergo ashing. The calculation of the tibia ash percentage in relation to the weight of the dry tibia was conducted to measure the ash content of the tibia without considering moisture. An inductively coupled plasma-optical emission spectrometer (ICP–OES; Optima 7000 DV, Perkin Elmer, Waltham, MA) was used to analyze the mineral content of the tibia ash and diet samples.

## Antibody response

The vaccination program involved administering vaccines for Newcastle disease virus (NDV) and infectious bronchitis virus (IBV). Blood samples were collected from broiler chickens (2 birds per replicate) on days 27 and 35, which corresponded to 7 and 14 days after the last vaccination, respectively. Serum samples were obtained through centrifugation, and the antibody titers were determined using a hemagglutination inhibition test [37] for NDV and ELISA kits for IBV [38]. The results are expressed as logarithmic values.

This study also examined the humoral immune response against sheep red blood cells (SRBC). On days 20 and 27, two birds in each replicate were immunized by injecting 0.25 mL of a solution containing 10% SRBC in phosphate-buffered saline (PBS). Blood samples (1.5 mL/bird) were collected from the brachial veins of the birds 7 days after each immunization (on days 27 and 35 after hatching). The blood samples were placed in serum tubes and left at room temperature for 2 h. Next, the samples were centrifuged at $1500 \times g$ for 10 minutes at 25˚C to separate the serum, which was later stored at -20˚C for antibody analysis. A direct hemagglutination assay using 96-well, U-bottomed microtiter plates was conducted to measure the serum antibody response against SRBC. Prior to the assay, the serum was heated at 56˚C for 30 minutes to inactivate it. The levels of total anti-SRBC antibodies and IgG antibodies resistant to mercaptoethanol were determined using the protocol outlined by Delhanty and Solomon [39]. The levels of IgM antibodies were determined by subtracting the IgG response from the total antibody response.

## Gene expression analysis

The RNA extraction and gene expression protocols used in this study were previously described by [40]. Total RNA was extracted from chicken liver samples using a commercial kit (Pars Tous Com., Iran), and the purity and quality of the RNA were assessed using a spectrophotometer (BioTek, USA). Genomic DNA was removed using DNase I (Thermo Fisher Scientific, Austin, TX, USA), and cDNA synthesis was performed using a commercial kit (Pars Tous, Iran). Gene expression levels were analyzed using quantitative polymerase chain reaction (qPCR) following the MIQE guidelines [41]. Various genes, including a reference gene [glyceraldehyde-3-phosphate (GAPDH)], immune-related genes [NF-kB, IL-1β, IFN-γ, IL-6, interleukin-10 (**IL-10**), and transforming growth factor beta (**TGF-β1**)], and antioxidant-related genes (Nrf2, GPx1, SOD1, and CAT), were examined. The reactions were performed in duplicate, with a total volume of 20 µl. SYBR Green qPCR Master Mix (Pars Tous, Iran) was used for RT–qPCR on an ABI 7300 Real Time PCR system (Applied Biosystems, Foster City, CA). The sequences of primers used in the study are presented in Table 2. The measurements were conducted in triplicate. The relative gene expression was determined using the $2^{-\Delta\Delta Ct}$ method, where the Ct values for the target gene were normalized to the reference gene GAPDH.

## Statistical analysis

The statistical analysis was conducted using the generalized linear model (GLM) procedure, following a completely randomized design. The analysis was performed using the SAS Institute software (version 9.0, SAS Institute, Cary, NC, USA). The Shapiro–Wilk test was used to evaluate the normality of the data, while the Levene test was used to investigate variance homogeneity. The pen served as the experimental unit for evaluating growth performance, while the individual bird was used to assess other physiological parameters. The LSMEANS option in SAS was utilized to evaluate the differences in means. To determine the significance of these

**Table 2. Gene special primers used in the real-time quantitative reverse transcription PCR.**

| Gene[1] | Primer sequence (5⁰–3⁰)[2] | Length (nt) | GenBank number |
|---|---|---|---|
| NF-kB | F: CCTGGCTGTTGTCGAATACCT | 154 | NM_001001472.3 |
| | R: CACTTTGTTCACATCTGCCCC | | |
| IFN-γ | F: TTCAGATGTAGCTGACGGTGG | 139 | NM_205149.2 |
| | R: CGGCTTTGACTTGTCAGTGTT | | |
| IL-6 | F: TTCAGCAATGGCAACAGCAATG | 156 | NM_204628.2 |
| | R: ATAGCAACAAGCGTCGTATTTCAAC | | |
| IL-10 | F: GCTCTCACACCGCCTTGC | 216 | NM_001004414.4 |
| | R: ACTGCTTAACTGCTATCACTAACTCTC | | |
| IL-1β | F: TCTTCTACCGCCTGGACAGC | 145 | XM_046931582.1 |
| | R: TAGGTGGCGATGTTGACCTG | | |
| TGF-β1 | F: GGAGTTATTTGGGGGGGGGT | 141 | NM_001318456.1 |
| | R: GCGTTTCTTTTTGGCCGCCTC | | |
| Nrf2 | F: GGCTTCTCCAGCTGCATTTC | 176 | XM_046921130.1 |
| | R: TACTTCAGCCAGGTTGTCGTT | | |
| GPx1 | F: CCATGTTCGAGAAGTGCGAGG | 120 | NM_001277853.3 |
| | R: TGTACTGCGGGTTGGTCATCA | | |
| SOD1 | F: AATCTCATTACTACTCTGCGTTCTTG | 121 | NM_205064.2 |
| | R: CCCAATCAACCATCTTCCATTACAC | | |
| CAT | F: GTTGGCGGTAGGAGTCTGGTCT | 182 | NM_001031215.1 |
| | R: GTGGTCAAGGCATCTGGCTTCTG | | |
| GAPDH | F: CAGAACATCATCCCAGCGTCCAC | 134 | NM_204305.2 |
| | R: CGGCAGGTCAGGTCAACAACAG | | |

[1]NF-kB, nuclear factor kappa B; IFN-γ, interferon gamma; IL-6, interleukin-6; IL-10, interleukin-10; IL-1β, interleukin-1 beta; TGF-β1, transforming growth factor-beta; Nrf2, nuclear factor erythroid 2-related factor 2; GPx1, glutathione peroxidase 1; SOD1, superoxide dismutase 1; CAT, catalase, and GAPDH, glyceraldehyde-3-phosphate.

[2]F = forward primer; R = reverse primer.

differences, Tukey's test was employed, with adjustments made at a significance level of $P < 0.05$.

## Results

### Growth performance

The performance of broiler chickens under HS conditions is presented in Fig 1. The body weight gain from 0 to 42 days in the ACTM100 and ITM+ACTM25 treatments was similar to that recorded in the TNC treatment group, but it was greater than that in the HSC and ACTM50 treatment groups ($P < 0.05$). Throughout the entire trial period, there was a significant increase in the feed intake of the TNC treatment group compared to that of all the other HS treatment groups ($P < 0.05$). Compared with the HSC, ACTM50, and ITM125 treatments, the ACTM100 treatment resulted in a significantly greater feed efficiency ($P < 0.05$). The overall feed efficiency also significantly increased in the ITM+ACTM25 and TNC treatment groups compared to that in the HSC treatment group ($P < 0.05$). Additionally, the birds in the TNC treatment group were more uniform than those in the HSC and ACTM50 treatment groups.

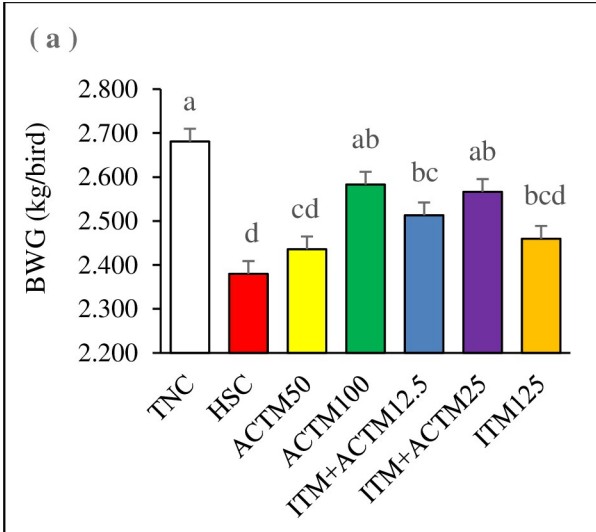
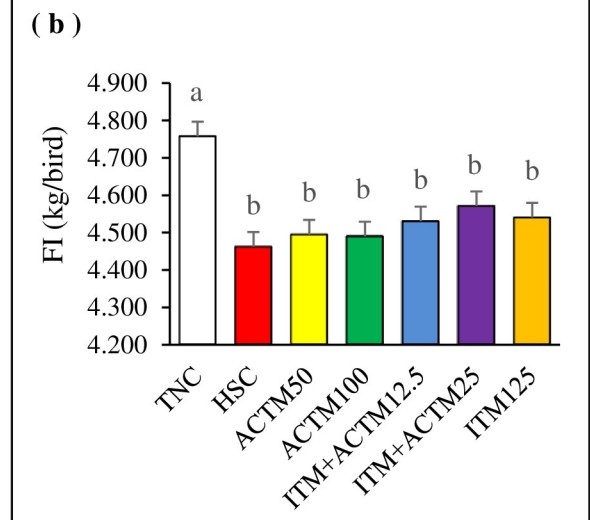
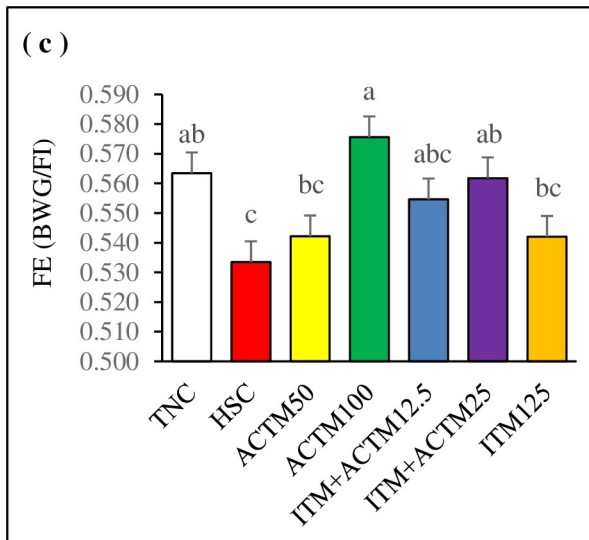
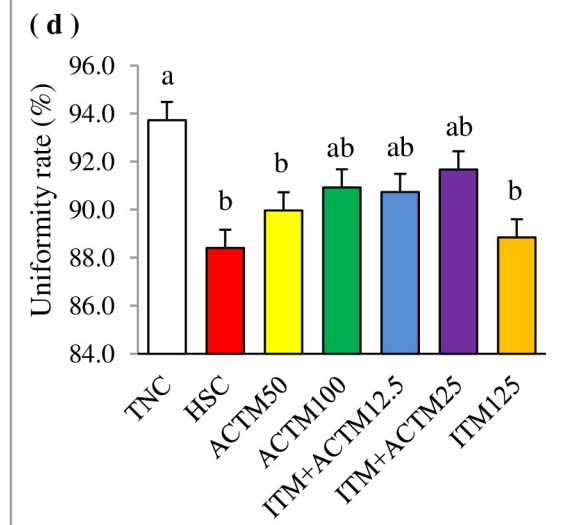

**Fig 1.** Bar charts of body weight gain (BWG; a), feed intake (FI; b), feed efficiency (FE; c), and uniformity rate (d) in heat-stressed broilers from 0 to 42 days of age. [a–c] Different letters in the same histogram indicate significant differences among groups according to Tukey's multiple range test ($P < 0.05$). Abbreviations: TNC, thermoneutral control group with commercially recommended levels of inorganic trace minerals (ITM). Heat stress (HS) groups included: HSC, heat stress control group with commercially recommended levels of ITM; ACTM50, advanced chelate technology-based trace minerals (ACTM) matching 50% of the ITM; ACTM100, ACTM equivalent to ITM; ITM+ACTM12.5, ACTM added to the ITM diet at 12.5% above recommended levels for TM; ITM+ACTM25, ACTM added to the ITM at 25% above recommended levels for TM; ITM125, 125% of commercially recommended levels of ITM.

## Bone characteristics

The results of the analysis of bone characteristics in heat-stressed broiler chickens at 42 days of age are presented in Table 3. Tibial length and weight were greater ($P < 0.05$) in the ACTM100 and TNC treatments than in the HSC and ACTM50 treatments. Compared with the HSC treatment group, the ITM+ACTM25 treatment group also exhibited greater ($P < 0.05$) tibia length and weight. A significant increase in the diameter of the tibia at its widest point was observed in the TNC treatment group compared to the HSC treatment group ($P < 0.05$). The ash and phosphorus levels in the tibia of birds receiving the ACTM100 and

**Table 3. Effect of advanced chelate technology-based trace minerals on bone morphology and mineral contents observed in broiler chickens challenged with heat stress.**

| Item | Experimental groups[1] | | | | | | | SEM | P-value |
|---|---|---|---|---|---|---|---|---|---|
| | TNC | HSC | ACTM50 | ACTM100 | ITM+ACTM12.5 | ITM+ACTM25 | ITM125 | | |
| length (mm) | 105.5[a] | 96.0[b] | 96.9[bc] | 104.3[a] | 99.8[abc] | 103.7[ab] | 99.6[abc] | 1.63 | <0.001 |
| weight (g) | 15.09[a] | 12.79[d] | 13.10[cd] | 14.50[ab] | 13.55[bcd] | 14.38[abc] | 13.41[bcd] | 0.307 | 0.0002 |
| Relative tibia weight | 0.530 | 0.506 | 0.513 | 0.548 | 0.514 | 0.546 | 0.518 | 0.0145 | 0.262 |
| Diameter (widest, mm) | 9.88[a] | 9.30[b] | 9.40[ab] | 9.73[ab] | 9.53[ab] | 9.73[ab] | 9.52[ab] | 0.113 | 0.009 |
| Diameter (narrowest, mm) | 7.59 | 7.15 | 7.19 | 7.44 | 7.24 | 7.38 | 7.24 | 0.1238 | 0.158 |
| Ash (%) | 52.44[a] | 48.41[b] | 48.90[ab] | 52.19[a] | 51.34[ab] | 52.09[a] | 50.12[ab] | 0.821 | 0.002 |
| Calcium (g/kg) | 166.3 | 158.8 | 160.3 | 163.0 | 159.2 | 165.9 | 159.1 | 2.44 | 0.1226 |
| Phosphorus (g/kg) | 95.9[a] | 88.2[b] | 90.7[ab] | 94.9[a] | 91.2[ab] | 94.8[a] | 90.8[ab] | 1.496 | 0.004 |
| Magnesium (g/kg) | 4.66 | 4.01 | 4.10 | 4.13 | 4.14 | 4.43 | 4.09 | 0.158 | 0.0643 |
| Potassium (g/kg) | 2.25 | 2.03 | 2.03 | 2.12 | 2.14 | 2.19 | 2.10 | 0.081 | 0.4395 |
| Sodium (g/kg) | 5.54[a] | 4.65[b] | 4.58[b] | 4.68[b] | 4.67[b] | 4.86[ab] | 4.76[ab] | 0.1867 | 0.0114 |
| Iron (mg/kg) | 131.8[ab] | 118.7[b] | 122.8[ab] | 138.3[a] | 128.3[ab] | 137.2[a] | 125.0[ab] | 4.08 | 0.0095 |
| Zinc (mg/kg) | 170.3[abc] | 162.0[bc] | 159.8[c] | 182.5[a] | 169.9[abc] | 178.6[ab] | 165.6[abc] | 3.89 | 0.0009 |
| Manganese (mg/kg) | 4.66[abc] | 3.70[c] | 4.06[bc] | 5.07[a] | 4.52[abc] | 4.78[ab] | 4.40[abc] | 0.222 | 0.0016 |
| Copper (mg/kg) | 0.819[ab] | 0.736[b] | 0.798[ab] | 0.853[ab] | 0.840[ab] | 0.897[a] | 0.852[ab] | 0.0287 | 0.0110 |

Means within a row not sharing the same superscript are different at $P < 0.05$. Values are means of 8 replicates (pens) per treatment.

[1]TNC: Thermoneutral control group with commercially recommended levels of inorganic trace minerals (ITM). Heat stress groups contained: HSC, heat stress control group with commercially recommended levels of ITM; ACTM50, advanced chelate technology-based trace minerals (ACTM) match to 50% of the ITM; ACTM100, ACTM equivalent to ITM; ITM+ACTM12.5, ACTM added to the ITM diet at the level of 12.5% above the commercially recommended levels for TM; ITM+ACTM25, ACTM added to the ITM at the level of 25% above the recommended levels for TM. ITM125, 125% of the commercially recommended levels of ITM.

ITM+ACTM25 treatments were similar ($P > 0.05$) to those observed in birds receiving the TNC treatment but greater ($P < 0.05$) than those in birds receiving the HSC treatment. TNC exhibited the highest tibia sodium concentration ($P < 0.05$), which was significantly different from that of the other treatments, except for the ITM+ACTM25 and ITM125 treatments. The Fe content in the tibia was greater ($P < 0.05$) in the ACTM100 and ITM+ACTM25 treatments than in the HSC treatment. Compared with the HSC and ACTM50 treatments, the ACTM100 treatment resulted in the greatest increase in the tibia Zn and Mn contents ($P < 0.05$), with a statistically significant difference observed. The tibia Zn content was also greater ($P < 0.05$) in the ITM+ACTM25 treatment group than in the ACTM50 treatment group. Moreover, the tibia Mn and Cu contents were greater ($P < 0.05$) in the ITM+ACTM25 treatment group than in the HSC treatment group. However, there was no significant change in tibial calcium, magnesium, or potassium content due to the experimental treatments ($P > 0.05$).

## Humoral immune response

The antibody responses to different antigens are provided in Table 4. Compared with those in the HSC group, the primary anti-NDV antibody titers and secondary anti-IBV antibody titers in the TNC, ACTM100, and ITM+ACTM25 groups were significantly greater (P < 0.05). The experimental treatments did not have any significant effects on the primary anti-SRBC antibody responses (P > 0.05). In contrast, the secondary total anti-SRBC titers were significantly greater (P < 0.05) in the TNC, ACTM100, and ITM+ACTM25 treatment groups than in the HSC treatment group. The secondary anti-SRBC IgM response was also significantly greater in the TNC treatment group than in the HSC and ACTM50 treatment groups (P < 0.05).

**Table 4. Effect of advanced chelate technology-based trace minerals on antibody responses against Newcastle disease virus (NDV), infectious bronchitis virus (IBV), and sheep red blood cells (SRBC) observed in broiler chickens challenged with heat stress.**

| Item | Experimental groups[1] | | | | | | | SEM | P-value |
|---|---|---|---|---|---|---|---|---|---|
| | TNC | HSC | ACTM50 | ACTM100 | ITM+ACTM12.5 | ITM+ACTM25 | ITM25 | | |
| **Anti-NDV titer[2] ($log_2$)** | | | | | | | | | |
| **day 27** | 3.13[a] | 2.50[b] | 2.72[ab] | 3.09[a] | 2.89[a] | 3.06[a] | 2.80[ab] | 0.125 | 0.012 |
| **day 35** | 2.84 | 2.20 | 2.50 | 2.74 | 2.57 | 2.74 | 2.52 | 0.139 | 0.056 |
| **Anti-IBV titer[3] ($log_{10}$)** | | | | | | | | | |
| **day 27** | 3.79 | 3.49 | 3.69 | 3.76 | 3.65 | 3.75 | 3.61 | 0.079 | 0.145 |
| **day 35** | 4.68[a] | 4.34[b] | 4.47[ab] | 4.60[a] | 4.48[ab] | 4.61[a] | 4.49[ab] | 0.069 | 0.037 |
| **Anti-SRBC antibody response[4] (log2)** | | | | | | | | | |
| **Primary** | | | | | | | | | |
| **Total antibody** | 2.53 | 2.05 | 2.42 | 2.44 | 2.43 | 2.41 | 2.61 | 0.125 | 0.098 |
| **IgG** | 1.63 | 0.72 | 1.23 | 1.23 | 1.12 | 1.43 | 1.35 | 0.252 | 0.289 |
| **IgM** | 1.35 | 1.23 | 1.52 | 1.38 | 1.42 | 1.32 | 1.82 | 0.259 | 0.776 |
| **Secondary** | | | | | | | | | |
| **Total antibody** | 3.20[a] | 2.70[c] | 2.82[bc] | 3.13[ab] | 2.99[abc] | 3.12[ab] | 2.91[abc] | 0.102 | 0.018 |
| **IgG** | 1.75 | 1.78 | 2.10 | 1.96 | 1.72 | 1.95 | 1.85 | 0.211 | 0.857 |
| **IgM** | 2.53[a] | 1.52[bc] | 1.13[c] | 2.14[ab] | 2.13[ab] | 2.03[abc] | 1.92[abc] | 0.295 | 0.050 |

Means within a row not sharing the same superscript are different at $P < 0.05$. Values are means of 8 replicates (pens) per treatment.

[1]TNC: Thermoneutral control group with commercially recommended levels of inorganic trace minerals (ITM). Heat stress groups contained: HSC, heat stress control group with commercially recommended levels of ITM; ACTM50, advanced chelate technology-based trace minerals (ACTM) match to 50% of the ITM; ACTM100, ACTM equivalent to ITM; ITM+ACTM12.5, ACTM added to the ITM diet at the level of 12.5% above the commercially recommended levels for TM; ITM+ACTM25, ACTM added to the ITM at the level of 25% above the recommended levels for TM. ITM125, 125% of the commercially recommended levels of ITM.

[2] Hemagglutinin inhibition test.

[3] The mean ELISA antibody response.

[4] Hemagglutinin inhibition test for primary (day 27) and secondary (day 35) antibody response against SRBC.

## Gene expression related to the NF-kB signaling pathway

The results pertaining to gene expression in the liver related to the NF-kB pathway are presented in Fig 2. The expression of the Nf-kB gene in the ACTM100 treatment group was similar to that in the TNC treatment group but lower than that in the HSC treatment group ($P < 0.05$). Decreases in IL-1β and IL-6 gene expression levels were observed in the ACTM100, ITM+ACTM25, and TNC treatment groups compared to the HSC treatment group ($P < 0.05$). The expression of the IFN-γ gene was greater ($P < 0.05$) in the ACTM100 and ITM+ACTM25 treatment groups than in the TNC treatment group but lower ($P < 0.05$) than in the other treatment groups, except for the ITM+ACTM12.5 treatment group. Furthermore, the expression of the hepatic TGF-β gene was greater ($P < 0.05$) in the ACTM100, ITM+ACTM25, and TNC treatment groups than in the other treatment groups, except for the ITM+ACTM12.5 treatment group. Notably, the TNC treatment group exhibited the highest level of TGF-β expression ($P < 0.05$).

## Blood antioxidant status

According to Table 5, the TAC and SOD activity were lower for the ACTM100 and ITM+-ACTM25 treatments than for the HSC treatment ($P < 0.05$). The GPx activity of birds receiving the ACTM100 and ITM+ACTM25 treatments was comparable ($P > 0.05$) to that of those receiving the TNC treatment but greater ($P < 0.05$) than that of those receiving the HSC and

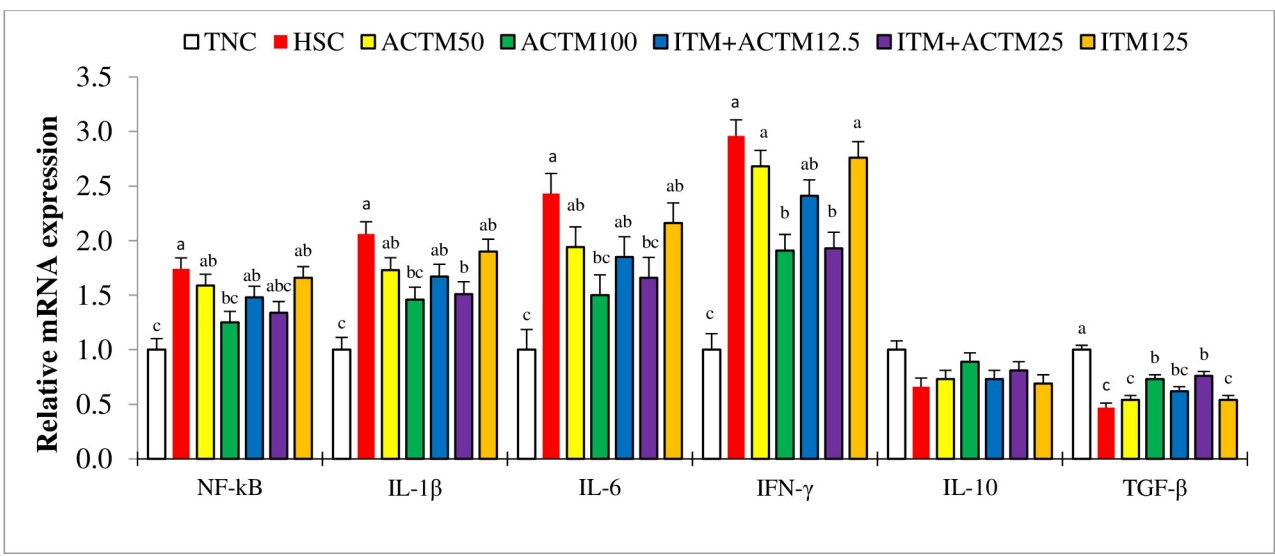

**Fig 2. Bar charts of hepatic mRNA expression levels of nuclear factor kappa B (NF-kB), interleukin (IL)-1β, IL-6, IL-10, interferon-γ (INF-γ), and transforming growth factor-β (TGF-β) in broilers at 42 days of age.** [a–c] Different letters in the same histogram indicate significant differences among groups according to Tukey's multiple range test ($P < 0.05$). Abbreviations: TNC, thermoneutral control group with commercially recommended levels of inorganic trace minerals (ITM). Heat stress (HS) groups included: HSC, heat stress control group with commercially recommended levels of ITM; ACTM50, advanced chelate technology-based trace minerals (ACTM) matching 50% of the ITM; ACTM100, ACTM equivalent to ITM; ITM+ACTM12.5, ACTM added to the ITM diet at 12.5% above recommended levels for TM; ITM+ACTM25, ACTM added to the ITM at 25% above recommended levels for TM; ITM125, 125% of commercially recommended levels of ITM.

ACTM50 treatments. Greater GPx activity was also detected in the ITM+ACTM12.5 and ITM125 treatments than in the HSC treatment ($P < 0.05$). The lowest serum MDA concentration was also observed in the TNC and ACTM100 treatments, which exhibited a significant difference from the HSC and ACTM50 treatments ($P < 0.05$). The serum MDA concentration in the ITM+ACTM25 treatment group was also lower ($P < 0.05$) than that in the HSC treatment group, while it was comparable to that in the TNC treatment group. However, the experimental treatments had no effect on serum CAT activity at 42 days of age ($P > 0.05$).

**Table 5. Effect of advanced chelate technology-based trace minerals on serum antioxidant status and hormone profile observed in broiler chickens challenged with heat stress.**

| Item | Experimental groups[1] | | | | | | | SEM | *P*-value |
|---|---|---|---|---|---|---|---|---|---|
| | TNC | HSC | ACTM50 | ACTM100 | ITM+ ACTM12.5 | ITM+ ACTM25 | ITM125 | | |
| **Total antioxidant capacity (U/mL)** | 3.86[ab] | 2.89[b] | 3.52[ab] | 4.11[a] | 3.61[ab] | 4.02[a] | 3.59[ab] | 0.2419 | 0.0187 |
| **Glutathione peroxidase (U/mL)** | 1883[a] | 1325[c] | 1441[bc] | 1930[a] | 1708[ab] | 1835[a] | 1673[ab] | 73.26 | < .0001 |
| **Superoxide dismutase (U/mL)** | 196.2[a] | 167.4[b] | 177.2[ab] | 218.1[a] | 182.8[ab] | 209.8[a] | 176.6[ab] | 9.73 | 0.0038 |
| **Catalase (U/mL)** | 2.54 | 2.22 | 2.41 | 2.61 | 2.43 | 2.76 | 2.33 | 0.1436 | 0.1760 |
| **Malondialdehyde (nmol/mL)** | 1.70[d] | 2.75[a] | 2.48[ab] | 1.87[cd] | 2.33[abc] | 1.94[bcd] | 2.42[abc] | 0.1358 | < .0001 |

Means within a row not sharing the same superscript are different at $P < 0.05$. Values are means of 8 replicates (pens) per treatment.

[1]TNC: Thermoneutral control group with commercially recommended levels of inorganic trace minerals (ITM). Heat stress groups contained: HSC, heat stress control group with commercially recommended levels of ITM; ACTM50, advanced chelate technology-based trace minerals (ACTM) match to 50% of the ITM; ACTM100, ACTM equivalent to ITM; ITM+ACTM12.5, ACTM added to the ITM diet at the level of 12.5% above the commercially recommended levels for TM; ITM+ACTM25, ACTM added to the ITM at the level of 25% above the recommended levels for TM. ITM125, 125% of the commercially recommended levels of ITM.

### Gene expression of Nrf2 signaling pathway components

Fig 3 displays the hepatic gene expression results for the Nrf2 pathway. Nrf2 gene expression was greater in all experimental treatments, except for the ITM+ACTM12.5 and ITM125 treatments, than in the HSC treatment. Additionally, there was greater expression of the SOD1 gene ($P < 0.05$) in the ACTM100 and TNC treatment groups than in the HSC treatment group. The expression of the GPx1 gene was also greater ($P < 0.05$) in the ACTM100, ITM+ACTM25, ITM+ACTM12.5, and TNC treatment groups than in the HSC treatment group. Among these treatments, the TNC treatment had the greatest effect on GPx1 gene expression ($P<0.05$). Additionally, a reduction in CAT gene expression was observed in the HSC and ACTM50 treatment groups compared to that in the TNC treatment group ($P < 0.05$).

## Discussion

The current study aligns with previous research indicating that broiler chickens have reduced feed intake when exposed to high temperatures [36,42]. Heat stress conditions may negatively impact feed intake by affecting the presence and distribution of enteroendocrine cells, which regulate appetite [43]. Reduced feed consumption also decreases the availability of nutrients for growth and development [44]. Heat stress can also increase corticosterone levels, which hinder broiler growth and development by impeding thyroid hormone synthesis and restricting feed intake [45]. Furthermore, Quinteiro-Filho et al. [46] reported that HS activates the hypothalamic–pituitary–adrenal axis, causing a redistribution of energy resources away from growth processes, resulting in decreased overall growth performance.

This study revealed that using advanced chelate sources of TM at 50% of the recommended commercial level can maintain growth performance in broiler chickens. Our findings also indicate that substituting ITM with ACTM or adding ACTM to the ITM diet at a dosage exceeding the recommended level of 25% could improve productive performance under HS

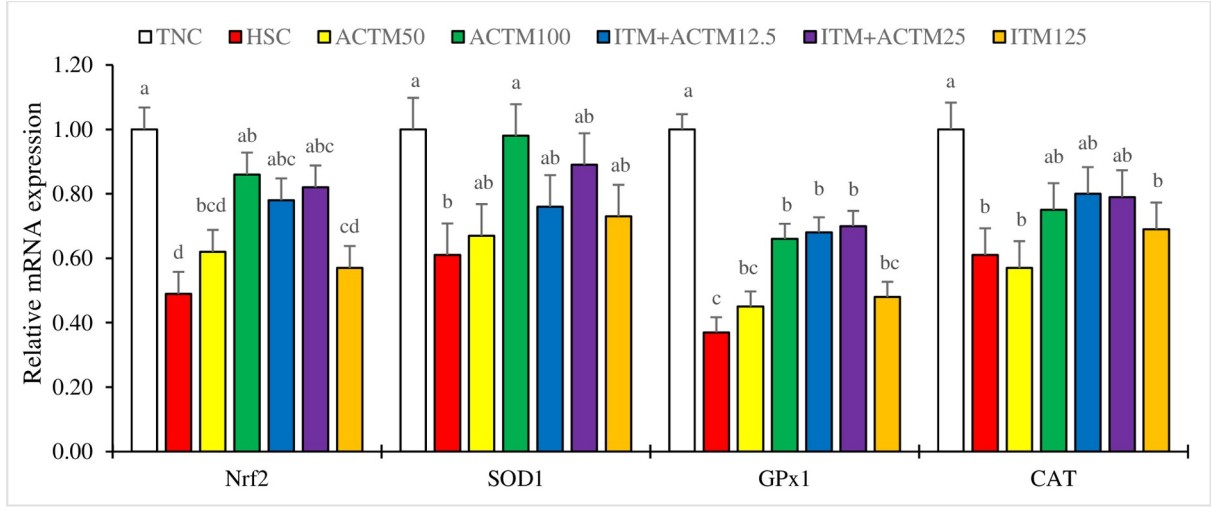

**Fig 3. Bar charts of hepatic mRNA expression levels of nuclear factor erythroid 2-related factor 2 (Nrf2), glutathione peroxidase 1 (GPx1), superoxide dismutase 1 (SOD1), and catalase (CAT) in broilers at 42 days of age.** [a–d] Different letters in the same histogram indicate significant differences among groups according to Tukey's multiple range test ($P < 0.05$). Abbreviations: TNC, thermoneutral control group with commercially recommended levels of inorganic trace minerals (ITM). Heat stress (HS) groups included: HSC, heat stress control group with commercially recommended levels of ITM; ACTM50, advanced chelate technology-based trace minerals (ACTM) matching 50% of the ITM; ACTM100, ACTM equivalent to ITM; ITM+ACTM12.5, ACTM added to the ITM diet at 12.5% above recommended levels for TM; ITM+ACTM25, ACTM added to the ITM at 25% above recommended levels for TM; ITM125, 125% of commercially recommended levels of ITM.

conditions. This finding aligns with previous research indicating that dietary supplementation with 500 ppm organic minerals can positively influence broiler growth [47]. Additionally, Rajkumar et al. [48] showed that incorporating organic forms of Se, Cr, and Zn into the diet can improve overall growth performance in heat-stressed broilers. Baxter et al. [11] also indicated that incorporating organic TM into the diet can reduce stress, improve the inflammatory response, enhance gut integrity, and ultimately improve growth performance. The enhanced growth performance in the ACTM100 and ITM+ACTM25 treatments may be due to improvements in immune system function and heightened activity of antioxidant enzymes, as indicated in this study. The sophisticated chelated structure of the organic acid-chelated TM supplement is believed to have a greater impact on improving growth performance in broiler chickens fed the ACTM100 and ITM+ACTM25 diets. The spatial arrangement of organic acid molecules in the coordination sphere of metal ions creates optimal conditions for chelating minerals, which promotes biological characteristics [33,34]. In addition, the advantages of advanced chelation technology-based TM may be partly attributed to the organic acids present in the supplement. The supplement's organic acids have several health effects after they penetrate enterocyte cells [29,35]. Several studies have shown that including organic acids in broilers' diets has positive effects on gut morphology, intestinal mucosal barrier function, and the number of harmful bacteria in the intestine [49–51]. Therefore, it seems that improving gut health through the use of organic acids may lead to better nutrient digestion and absorption, which in turn might boost growth performance.

Based on the results of the present study, exposure to HS had detrimental impacts on various aspects of the tibia bone, including its morphological characteristics (length, weight, and diameter), as well as its ash, phosphorus, and sodium contents. In previous studies, it was observed that HS has negative effects on bone quality parameters in broilers [52] and pullets [53]. Heat stress can have detrimental effects on bone health through various mechanisms. These include disruptions in the gastrointestinal tract, which can affect the absorption of important nutrients such as phosphorus [54]. In addition, HS has the potential to compromise the immune system, which can result in the transportation of pathogens and their byproducts to bones via the bloodstream, leading to inflammation and damage [55]. The functioning of the hypothalamic–pituitary–adrenal axis can also be affected, leading to negative impacts on bone remodeling and increased cell death [56]. Furthermore, HS can impair the antioxidant system of bone cells, resulting in oxidative stress [57].

The best results in terms of bone morphology and higher levels of ash, phosphorus, and TM were observed when ITM was completely replaced with ACTM in the ACTM100 treatment or when ACTM was added to the ITM diet at an extra level of 25% of the recommended dosage in the ITM+ACTM25 treatment. These findings suggest that the heat-stressed broilers in these groups had more bone components available. Under typical environmental conditions, comparable results were observed in terms of the physical characteristics and mineral profile of the tibia bone when the inorganic TM premix was completely replaced with advanced chelate technology-based TM in the diet of broilers [35] and turkeys [34]. As previously stated, the presence of HS leads to oxidative stress and inflammation in bone cells [57,58]. Therefore, one potential reason for the improvement of bone characteristics in the ACTM100 and ITM+ACTM25 treatments could be linked to the observed enhancement in antioxidant status and anti-inflammatory properties in this study. The elevated levels of ash and phosphorous in the tibia of broilers treated with ACTM100 and ITM+ACTM25 also support the concept that organic TM supplementation may reduce the interference caused by substances (particularly phytate) that form insoluble complexes with TM [27,35].

There is a clear connection between the prolonged impact of stress on the hypothalamic–pituitary–adrenal axis and a decrease in immune system functioning [59,60]. In addition, HS

causes oxidative stress, resulting in the oxidation of lipids in the cell membrane of immune-related cells, which can lead to a decrease in immune responses [45,61]. Our findings revealed a noticeable decline in humoral immune responses, specifically in terms of antibody titers against various antigens (NDV, IBV, and SRBC), in broiler chickens raised under HS conditions. The transcription factor NF-kB plays a crucial role in regulating genes involved in the immune response and inflammation [62,63]. Activation of the NF-kB signaling pathway has been shown to elicit the synthesis and release of proinflammatory cytokines, thereby playing a crucial role in the initiation and progression of the inflammatory response [6]. Once produced, these proinflammatory cytokines actively participate in the inflammatory process, further exacerbating the immune response [64]. Previous studies have demonstrated a significant increase in NF-kB protein expression in response to HS [5,7]. Our findings also revealed that prolonged exposure to HS resulted in a significant upregulation of NF-kB expression and subsequent upregulation of IFN-γ-, IL-1β-, and IL-6-related proteins in the liver. These results indicate that the activation of the NF-kB signaling pathway and subsequent secretion of the inflammatory factors IFN-γ, IL-1β, and IL-6 are key mechanisms by which HS induces inflammation in the livers of broilers.

The results of the current study indicate that completely substituting inorganic TM sources with advanced chelated TM has the potential to offer protection against HS conditions by effectively regulating crucial immune-related genes. The implementation of this replacement strategy resulted in the downregulation of NF-kB, a decrease in the expression of the proinflammatory cytokines IL-1β, IL-6, and IFN-γ, and an increase in the expression of the anti-inflammatory cytokine TGF-β. The incorporation of ACTM in ITM diets at 12.5% and 25% (referred to as the ITM+ACTM12.5 and ITM+ACTM25 treatments, respectively) has also been shown to regulate the immune response by reducing the activity of NF-kB and IFN-γ. Our study also revealed that the ACTM100 and ITM+ACTM25 treatments increased the serum antibody titers against NDV, IBV, and SBRC at different ages in heat-stressed broilers, suggesting a positive effect on humoral immune function. The use of ACTM seems to reduce the immunopathological effects caused by HS, resulting in decreased hepatic inflammation and systemic inflammation. In accordance with our findings, it has been observed that the inclusion of an organic form of zinc (Zn–Gly chelate), as opposed to zinc sulfate, as a dietary supplement for broiler chickens has the potential to modulate the immune response [24]. This modulation is achieved through the reduction of proinflammatory cytokine production and the enhancement of the expression of anti-inflammatory factors, particularly IL-10. A previous study also indicated that the replacement of ITM (Fe, Cu, Mn, and Zn in the form of sulfates and Se as sodium selenite) with organic TM led to an increase in blood IgG levels [22]. In a previous investigation conducted on laying hens exposed to elevated ambient temperatures, it was observed that the inclusion of dietary organic mineral mixture supplemented at levels of 0.5 and 1 g/kg resulted in enhanced antibody titers against avian influenza H9N1 and NDV compared to the control diet [65]. The addition of an organic acid-TM chelate supplement in this study has the potential to enhance the absorption of TM, thereby increasing its availability for various physiological processes, such as supporting optimal immune system functions. By ensuring an adequate supply of TM, the immune system can function at its optimal efficiency, thereby enhancing its ability to combat difficult conditions and maintain overall health [66]. In addition, the observed increase may be partly due to the release of organic acids in the enterocytes of the gastrointestinal tract following the administration of this supplement. Based on a previous investigation, the inclusion of organic acids in poultry feed was shown to improve both humoral and cell-mediated immune responses while also effectively modulating cytokine levels within the host organism [67].

Heat stress disrupts the balance between cellular antioxidants and oxidants, which in turn affects the homeostasis of these crucial components inside the cell [68]. The transcription factor Nrf2 is recognized as a crucial regulator of antioxidant capacity in the body. The significance of this phenomenon lies in its ability to modulate antioxidant defense mechanisms in various organs, such as the intestine [69], liver [9], and kidneys [70]. A recent study revealed a decrease in the expression of Nrf2 in hepatocytes exposed to HS [9]. The findings of this study revealed a significant reduction in the hepatic mRNA levels of Nrf2, GPx1, and SOD1 due to HS. In addition, HS significantly decreased the levels of key antioxidant enzymes, including GPX, SOD, and CAT. The results obtained from this study also revealed that exposure to HS led to a significant increase in TAC while simultaneously elevating the serum concentration of MDA.

The examination of gene expression patterns associated with antioxidation revealed that replacing ITM with ACTM completely or supplementing the ITM diet with ACTM at a dosage 12.5 or 25% higher than the recommended level has been found to enhance the hepatic expression of the Nrf2 gene, which is responsible for producing antioxidant enzymes, especially hepatic SOD1 and GPx1. Furthermore, the ACTM100 and ITM+ACTM25 treatments led to increased levels of serum TAC, GPx, and SOD but simultaneously decreased levels of serum MDA in response to the HS challenge. The present study proposes that incorporating dietary supplementation with ACTM, especially in the ACTM100 and ITM+ACTM25 groups, could mitigate the negative effects of HS on broiler chickens and may help the chickens achieve a stable physiological state similar to that of the nonchallenged group. Consistent with the findings of this study, the results of a recent study indicated that replacing ITM with organic TM, which contained 50 ppm Fe, 30 ppm Zn, 15 ppm Mn, and 0.2 ppm Se or more, resulted in a significant increase in the gene expression of SOD and GPx in the ileum of piglets [71]. In another study on broiler breeders, Umar Yaqoob et al. [23] discovered that substituting commercial levels of ITM (Cu, Zn, Fe, and Mn sulfates) with 70% commercial levels of TM in the form of complexed glycinates had a positive impact on liver antioxidant status. The effectiveness of antioxidant stress defense relies on the presence of antioxidant minerals as cofactors. Specifically, Zn, Cu, and Mn function as cofactors for SOD, whereas Fe is a constituent of CAT, and Se acts as a cofactor for GPx. Hence, the presence of these minerals in sufficient amounts is essential for the effective operation of antioxidant stress defense mechanisms [34,35]. Consistent with previous research findings, it has been observed that organic forms of Zn, Mn, Cu, and Fe have the potential to enhance the production of SOD and CAT [72,73]. Furthermore, it has been discovered that organic Se can enhance the activity of GPx [74,75]. These results suggest that the inclusion of advanced chelate technology-based TM, particularly in the ACTM100 and ITM+ACTM25 diets, may have led to an increase in mineral bioavailability. This, in turn, could have contributed to a decrease in ROS accumulation and an improvement in the antioxidant defense system when compared to the HSC diet. However, additional research is necessary to investigate the underlying mechanisms and establish the ideal levels of TM inclusion for maximizing these benefits in broiler production systems under challenging conditions, as applied in this study.

In conclusion, ACTM supplementation has positive effects on productive performance, bone characteristics, and health status in heat-stressed broilers, especially when it is used as a complete replacement for inorganic TM or at a supplemental level of 25% in the feed. These findings also suggested that ACTM supplementation can protect broilers from heat stress by reducing the expression of inflammatory markers and increasing the expression of antioxidant proteins. These results suggest that further research is needed to understand the mechanism of action of ACTM and determine the optimum TM sources and requirements in broiler chickens reared under HS conditions.

## Supporting information

**S1 Fig. The BET analysis of advanced chelates crystallinity used in this study.**
(DOCX)

**S1 Table. Supplemental level and analyzed content of trace minerals in experimental diets (mg/kg).**
(DOCX)

**S1 Data.**
(XLSX)

## Author Contributions

**Conceptualization:** Taher Mohammadizad, Kamran Taherpour, Hossein Ali Ghasemi.

**Data curation:** Taher Mohammadizad, Hassan Shirzadi, Fatemeh Tavakolinasab.

**Formal analysis:** Kamran Taherpour, Hossein Ali Ghasemi.

**Funding acquisition:** Kamran Taherpour.

**Investigation:** Taher Mohammadizad, Kamran Taherpour, Fatemeh Tavakolinasab.

**Methodology:** Hossein Ali Ghasemi, Hassan Shirzadi, Fatemeh Tavakolinasab.

**Project administration:** Kamran Taherpour, Hossein Ali Ghasemi.

**Resources:** Kamran Taherpour, Hassan Shirzadi.

**Supervision:** Kamran Taherpour, Hossein Ali Ghasemi, Mohammad Hassan Nazaran.

**Validation:** Kamran Taherpour, Hassan Shirzadi.

**Visualization:** Taher Mohammadizad, Fatemeh Tavakolinasab.

**Writing – original draft:** Taher Mohammadizad, Fatemeh Tavakolinasab.

**Writing – review & editing:** Kamran Taherpour, Hossein Ali Ghasemi, Hassan Shirzadi, Mohammad Hassan Nazaran.

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
