## [Decision Letter · Decision Letter 0]

20 Aug 2024

PONE-D-24-23790Potential benefits of advanced chelate-based trace minerals in improving bone mineralization, oxidative status, humoral immunity, and gene expression modulation associated with antioxidation and immunity in heat-stressed broilersPLOS ONE

Dear Dr. Taherpour,

Thank you for submitting your manuscript to PLOS ONE. After careful consideration, we feel that it has merit but does not fully meet PLOS ONE’s publication criteria as it currently stands. Therefore, we invite you to submit a revised version of the manuscript that addresses the points raised during the review process.

We look forward to receiving your revised manuscript.

Kind regards,

Ewa Tomaszewska, DVM Ph.D

Academic Editor

PLOS ONE

A clean copy of the edited manuscript (uploaded as the new *manuscript* file)”.

3. Please include captions for your all Supporting Information files at the end of your manuscript, and update any in-text citations to match accordingly. Please see our Supporting Information guidelines for more information: http://journals.plos.org/plosone/s/supporting-information. 

Reviewers' comments:

Reviewer's Responses to Questions

**Comments to the Author**

1. Is the manuscript technically sound, and do the data support the conclusions?

Reviewer #1: Yes

2. Has the statistical analysis been performed appropriately and rigorously? 

Reviewer #1: Yes

3. Have the authors made all data underlying the findings in their manuscript fully available?

Reviewer #1: Yes

4. Is the manuscript presented in an intelligible fashion and written in standard English?

Reviewer #1: Yes

5. Review Comments to the Author

Reviewer #1: The authors address an important on heat stress in poultry

The manuscript is generally well written

A few comments to address

1 The title is so wordy and complicated. Can the authors simplify the title and keep the words to about twenty

The abstract requires an introductory statement on the topic of study instead of the “42 day experiment”

Line 231 what is the reference gene?

What are the limitations of the study?

What is the strength of the study?

6. PLOS authors have the option to publish the peer review history of their article (what does this mean?). If published, this will include your full peer review and any attached files.

Reviewer #1: **Yes: **ESTER LILIAN ACEN

---

## [Author Response · Author response to Decision Letter 0]

26 Aug 2024

Dear Editor, “PLOS ONE” Journal 

Thank you very much for giving us an opportunity to revise our manuscript (PONE-D-24-23790). Following the reviewer's suggestion, we have shortened the paper's title to "Potential benefits of advanced chelate-based trace minerals in improving bone mineralization, antioxidant status, immunity, and gene expression modulation in heat-stressed broilers." We responded to the reviewer's comments individually, indicating exactly how we addressed each comment and describing the changes we made. The authors want to extend their appreciation for taking the time and effort necessary to provide such insightful guidance. Additionally, the points offered by the journal have been carefully considered. All authors have approved the revisions and used MS Word's "Track Changes" option to mark changes related to reviewer comments and structural changes. Formatting changes in accordance with the journal guidelines and the "PLOS ONE style templates" have been marked with yellow highlighting to better differentiate them. We hope that this revision improves the paper so that you and the reviewer deem it worthy of publication.

Reviewer #1: The authors address an important on heat stress in poultry. The manuscript is generally well written.

A few comments to address

Response:

Dear Reviewer,

Thank you very much for your valuable comments on our manuscript. The authors have tried to be responsive to your comments and also responded to your comments. The lines mentioned in the parentheses are related to the highlighted version of the revised manuscript (with track change). 

1 - The title is so wordy and complicated. Can the authors simplify the title and keep the words to about twenty

Response: According to this comment, the length of the tile has been reduced to 21 words (L 4-7).

2- The abstract requires an introductory statement on the topic of study instead of the “42 day experiment”

Response: Thank you for this comment. The following sentence has been added at the beginning of the abstract: (L 21-23)

“Organic sources of trace minerals (TM) in broiler diets are more bioavailable and stable than inorganic sources, making them particularly beneficial during challenging periods such as heat stress (HS) conditions.“

3- Line 231 what is the reference gene?

Response: Addes as requested (L 225).

4- What are the limitations of the study?

Response: Thank you for bringing up this important query. Insufficient funding is indeed a major limitation in conducting research, especially when it comes to exploring cellular and molecular details. In these areas, the higher costs associated with laboratory work often limit the scope of research. In this particular study, the researchers attempted to examine immune and antioxidant responses by analyzing the genes involved in active pathways. However, a more comprehensive analysis of all genes involved in these cases, as well as the examination of multiple tissues, would require additional funding. Additionally, conducting challenging studies like heat stress with complex instructions and following study protocols can be challenging and costly, which is a common limitation in these kinds of studies.

5- What is the strength of the study?

Response: The strength of this study lies in its investigation of an organic trace mineral supplement using modern methods as a substitute for inorganic products. The study explores different replacement levels of inorganic supplements with organic supplements, as well as higher dietary supplemental levels of these supplements under stressful conditions. This is important because in commercial settings, feed is typically formulated with recommended levels of inorganic supplements. By comparing the effects of different dietary levels of organic supplements to the commercial levels of inorganic supplements, the study provides valuable insights into the potential benefits of using advanced chelate technology in improving bird performance and health in heat-stressed broilers. Additionally, the study uses a variety of measures to evaluate bone health, immune function, and oxidant/antioxidant status, which further strengthens its findings. Overall, this study highlights the potential benefits of advanced chelate technology in improving bird performance and health in heat-stressed broilers.

Again, we appreciate all of your insightful comments. We hope our revision meet your approval.

Best Regards,

Authors

---

## [Decision Letter · Decision Letter 1]

12 Sep 2024

Potential benefits of advanced chelate-based trace minerals in improving bone mineralization, antioxidant status, immunity, and gene expression modulation in heat-stressed broilers

PONE-D-24-23790R1

Dear Dr. Kamran Taherpour,

We’re pleased to inform you that your manuscript has been judged scientifically suitable for publication and will be formally accepted for publication once it meets all outstanding technical requirements.

Kind regards,

Ewa Tomaszewska, DVM Ph.D

Academic Editor

PLOS ONE

Additional Editor Comments (optional):

Reviewers' comments:

Reviewer's Responses to Questions

**Comments to the Author**

1. If the authors have adequately addressed your comments raised in a previous round of review and you feel that this manuscript is now acceptable for publication, you may indicate that here to bypass the “Comments to the Author” section, enter your conflict of interest statement in the “Confidential to Editor” section, and submit your "Accept" recommendation.

Reviewer #1: All comments have been addressed

2. Is the manuscript technically sound, and do the data support the conclusions?

Reviewer #1: Yes

3. Has the statistical analysis been performed appropriately and rigorously? 

Reviewer #1: Yes

4. Have the authors made all data underlying the findings in their manuscript fully available?

Reviewer #1: Yes

5. Is the manuscript presented in an intelligible fashion and written in standard English?

Reviewer #1: Yes

6. Review Comments to the Author

Reviewer #1: the authors need to include the strength and limitations of the study they presented in the rebuttal letter in the discussion section

7. PLOS authors have the option to publish the peer review history of their article (what does this mean?). If published, this will include your full peer review and any attached files.

Reviewer #1: **Yes: **ESTER LILIAN ACEN

---

## [Editor Report · Acceptance letter]

23 Sep 2024

PONE-D-24-23790R1 

PLOS ONE

Dear Dr. Taherpour, 

I'm pleased to inform you that your manuscript has been deemed suitable for publication in PLOS ONE. Congratulations! Your manuscript is now being handed over to our production team.

Kind regards, 

on behalf of

Professor Ewa Tomaszewska 

Academic Editor

PLOS ONE